# The Role of Tocotrienol in Preventing Male Osteoporosis—A Review of Current Evidence

**DOI:** 10.3390/ijms20061355

**Published:** 2019-03-18

**Authors:** Kok-Yong Chin, Soelaiman Ima-Nirwana

**Affiliations:** Department of Pharmacology, Faculty of Medicine, Universiti Kebangsaan Malaysia, Jalan Yaacob Latif, Bandar Tun Razak, 56000 Cheras, Malaysia; imasoel@ppukm.ukm.edu.my

**Keywords:** antioxidant, inflammation, men, mevalonate, osteopenia, skeleton, tocochromanol, vitamin E

## Abstract

Male osteoporosis is a significant but undetermined healthcare problem. Men suffer from a higher mortality rate post-fracture than women and they are marginalized in osteoporosis treatment. The current prophylactic agents for osteoporosis are limited. Functional food components such as tocotrienol may be an alternative option for osteoporosis prevention in men. This paper aims to review the current evidence regarding the skeletal effects of tocotrienol in animal models of male osteoporosis and its potential antiosteoporotic mechanism. The efficacy of tocotrienol of various sources (single isoform, palm and annatto vitamin E mixture) had been tested in animal models of bone loss induced by testosterone deficiency (orchidectomy and buserelin), metabolic syndrome, nicotine, alcoholism, and glucocorticoid. The treated animals showed improvements ranging from bone microstructural indices, histomorphometric indices, calcium content, and mechanical strength. The bone-sparing effects of tocotrienol may be exerted through its antioxidant, anti-inflammatory, and mevalonate-suppressive pathways. However, information pertaining to its mechanism of actions is superficial and warrants further studies. As a conclusion, tocotrienol could serve as a functional food component to prevent male osteoporosis, but its application requires validation from a clinical trial in men.

## 1. Introduction

Osteoporosis, a metabolic skeletal disease reflected by decreased bone mass, microarchitectural deterioration, and impaired bone strength, affects both men and women. The ultimate consequence of osteoporosis is bone fracture [1]. Women are more susceptible to bone fracture compared to men owing to the difference in bone strength and the presence of menopause in women [2]. However, bone fractures in men, which constitute 29% of all fragility fractures occurring worldwide, still pose a significant healthcare burden to society [3]. Of the 16.9 billion USD medical cost related to fracture, 4.15 billion USD was contributed by men [4]. Besides, men suffer from greater morbidity and mortality post-fracture compared to women [5,6]. They are also less likely to receive osteoporosis treatment compared women after a bone fracture [7].

Male osteoporosis can be classified into primary and secondary osteoporosis. The cause of primary male osteoporosis is age-related bone loss (senile osteoporosis) or unknown (idiopathic osteoporosis). Secondary male osteoporosis is caused by lifestyles, medical conditions, or medications harmful to the bone. Some of the lifestyle behaviors contributing to male osteoporosis are excessive consumption of alcohol and caffeinated beverages, cigarette-smoking, and physical inactivity. Male osteoporosis can occur secondary to other diseases, such as hypogonadism, gastrointestinal disease, hyperparathyroidism, and thyrotoxicosis. Prolonged use for medications, such as glucocorticoids, antineoplastic agents, and anticonvulsants, will also cause osteoporosis [8,9,10]. The underlying pathology of osteoporosis is imbalanced bone turnover, in which the bone resorption rate (mediated by osteoclasts) exceeds the bone formation rate (mediated by osteoblasts). The reasons for this imbalance vary, but it mainly stems from the disturbance of calcium homeostasis, hormonal changes, chronic inflammation and increased oxidative stress resulted from the risk factors aforementioned [11,12]. 

The current pharmaceutical interventions for osteoporosis are mainly targeted at postmenopausal women [2]. The currently approved drugs for male osteoporosis include bisphosphonates, teriparatide and denosumab [13]. However, studies revealed that most bone fractures occur in patients with osteopenia rather than those with osteoporosis [14]. The current prophylactic agent for osteoporosis is limited to calcium with or without vitamin D. Thus, functional food components may be a viable option to prevent deterioration of bone health. 

Tocotrienol (T3) is one of the functional food components being intensively investigated for its bone-sparing properties [15,16,17]. T3 and tocopherol (TP) belong to the family of tocochromanol (vitamin E), which is chemically characterized by a chromanol ring and a long carbon tail. Three carbon bonds at the position of 3, 7, 11 on the carbon tail are double bonds for T3, in contrast to single bonds for TP (Figure 1). This property enables T3 to integrate with the lipid bilayer and recycle free radicals better, thus explaining its superior antioxidant effect. T3 also possesses suppressive effects on the mevalonate pathway responsible in cholesterol production, a property not observed with TP. There are four different isoforms of T3, i.e., α, β, γ and δ-T3, depending on the number and position of the methyl group on the chromanol ring [18]. Natural tocochromanols usually exist in mixtures of varying composition in natural sources, such as botanical oil from palm kernel, annatto seed, rice bran, barley and wheat [19]. α-TP is the most abundant vitamin E in food and in the human body after supplementation with palm vitamin E supplementation, followed by α-T3 [20]. This is related to the binding of tocochromanols on α-TP transfer protein, which dictates their bioavailability in the blood [21].

Several broad reviews on the bone-protective effects of T3 have been published previously [15,17,22]. Briefly, T3 showed promising bone-sparing effects on animal models of bone loss due to estrogen deficiency induced by bilateral ovariectomy [23,24,25]. T3 was able to prevent deterioration of skeletal microarchitecture, mineral density, strength and calcium content [26,27,28]. The issues on the mechanism of bone protection by T3, whether through decreasing bone resorption or increasing bone formation or both, is still debatable [15]. Recent studies highlighted that T3 also protect male osteoporosis models [29,30]. Of note, T3 prevented gonadotropin-releasing hormone agonist (GnRH) [30,31] and metabolic syndrome-induced bone loss [32,33,34]. In view of the limited options of prophylactic agents for male osteoporosis, T3 may aid in the prevention of bone loss in high-risk men. This review aimed to provide an updated summary of the antiosteoporotic effects of T3 in male osteoporosis. Some of the causes of male osteoporosis will be elaborated further in this review because they are related to the bone loss models used to test T3. Since most of the studies are preclinical, the following discourse will be divided according to the models of bone loss in which the efficacy of T3 has been tested.

## 2. The Composition of T3 Used

The T3 investigated was usually extracted from natural sources, such as palm and annatto oil with varying composition of vitamin E isomers (Table 1). They were called palm vitamin E [33,35,36,37,38,39], palm/annatto T3 [29,30,31,32,40,41], T3 enriched fraction [42,43]. In some studies, pure T3 isomers were also used [42,44,45]. 

## 3. Effects of T3 on Bone Growth

Peak bone mass in humans is achieved during the third decade of life. A higher peak bone mass is protective against fragility fracture later in life. Apart from genetic factors, physical exercise, nutrition and health conditions may all play a role in determining the individual peak bone mass [47]. Studies were conducted to compare the effects of T3 isomers and αTP on the skeleton of normal male rats [44,45]. γT3 (oral, 60 mg/kg body weight (bw)/day, 4 months) was found to improve the bone structural, cellular and dynamic parameters better than α-TP at the same dose in normal male rats [45]. γT3 also performed better than δT3 (oral, 60 mg/kg bw/day, 4 months) in improving the cellular and dynamic histomorphometric indices in these rats [44]. Ultimately, rats treated with γT3 had greater bone strength than those treated with αTP [45]. 

On the other hand, an earlier study demonstrated that normal male rats (3 months old) treated with palm T3 (oral, 30 mg/kg bw/day, 8 months) did not change bone mass density (BMD) in any bone sites. However, it reduced the level of serum tartrate acid phosphatase (TRAP), a marker of bone resorption in these animals [35]. Since the dose used in this study was half of the previous studies [44,45], it might not be sufficient to promote growth. In another study, supplementation of high-dose palm T3 (500 mg/kg diet) for 18 weeks in normal male rats (10 weeks old) also did not alter the bone microstructure, BMD, bone mineral content, mineral apposition and bone formation rate [48]. The study also revealed that expression of genes associated with osteoblast differentiation, number and activity was not significantly altered by high-dose T3 supplementation [48]. The skeletal effects of high-dose T3 and α-TP were comparable [48]. The discrepancy in the skeletal effects of high-dose and low-dose T3 supplementation reflected that the bone anabolic effects of T3 diminish at high-dose, indicating a potential U-shaped relationship between bone and vitamin E, which has been hypothesized earlier [49]. At high-dose, T3 was reported to be toxic to bone cells [50]. 

Another study supplemented normal male rats using palm T3 mixture (oral, 100 mg/kg bw/day, 4 months) and showed evidence of decreased oxidative stress in the bone, marked by increased glutathione peroxidase and decreased lipid peroxidation product (malondialdehyde) [46]. αTP at the same dose did not change the level of both indicators in the bone [46]. Since free radicals induce osteoclast differentiation and oxidative damage on osteoblasts [51,52,53], the bone anabolic effects of the T3 could be attributed to its antioxidative activity. 

## 4. Effects of T3 on Bone Loss Due to Androgen Deficiency (Late-Onset or Drug-Induced Hypogonadism)

Androgen plays an important role in maintaining the bone health of men [54,55]. During bone growth, androgen is responsible for periosteal apposition in men. Androgen promotes the proliferation and differentiation of pre-osteoblasts [56]. It also enhanced the survival of osteoblasts [57]. Androgen deficiency induced through orchidectomy increases the proliferation of osteoclast and promotes bone resorption through increasing receptor activator of nuclear factor kappa-Β ligand (RANKL) synthesis [58]. Androgen can be transformed into estrogen via the aromatase enzyme and part of its skeletal effects are exerted through interaction with estrogen receptors [59]. This is clearly demonstrated by men with dysfunctional estrogen receptors or aromatase who are reported to suffer from severe osteoporosis [60,61,62,63]. 

Elderly men are susceptible to late-onset hypogonadism (LOH) and its complications. LOH is caused by dysregulation of the hypothalamic-gonadal axis, degeneration of the Leydig cells in testes and reduced bioavailability of testosterone due to increased sex-hormone binding globulin (SHBG) in ageing men [64]. However, unlike the menopause in women, testosterone deficiency is not universal in elderly men [65]. Epidemiological studies showed that age-related decline of free and bioavailable testosterone is associated with bone loss [66,67]. Besides, men receiving gonadotropin-releasing hormone (GnRH) agonist (a form of androgen ablation therapy) for prostate cancer are also susceptible to osteoporosis and bone loss [68]. 

Palm vitamin E mixture rich in T3 (oral, 30 m/kg bw/day, 8 months) was shown to prevent the decline in BMD and lumbar calcium content in orchidectomized rats [35]. Compared with rats fed with diet mixed with palm olein, serum alkaline phosphatase (ALP), a bone formation marker of rats supplemented with palm vitamin E was reduced, probably reflecting a reduced bone turnover rate [35]. The efficacy of annatto T3 mixture rich in δ-T3 (oral, 60 mg/kg bw/day, 8 weeks) was compared with testosterone replacement (intramuscular, 7 mg/kg weekly, 8 weeks) in orchidectomized rats. Annatto T3 reduced trabecular separation at the proximal tibia and the distal femur, as well as increased bone volume and trabecular number at the distal femur [29]. It also increased the calcein double-labelled surface of these rats but did not alter mineralizing surface and bone formation rate, indicating the increase in bone formation was marginal [29]. Osteoblast number increased, and osteoclast number was reduced in supplemented orchidectomized rats, but these were not reflected in the bone remodeling markers [29,40]. Their serum total calcium level was reduced and tibial calcium content increased [41]. However, this did not translate to an improvement in bone mechanical strength, indicating the changes at the microscopic level and mineral content was not enough to impact bone strength [41]. The authors postulated that given sufficient dose or treatment period, more promising results might be observed [41]. Mechanistically, mRNA expression of markers related to osteoblasts (ALP, collagen I α 1, β-catenin and osteopontin) was up-regulated prominently but there were no changes in markers related to osteoclasts in this study, showing that the action of annatto T3 may be more anabolic than anti-resorptive [29]. In contrast, the testosterone supplemented group showed significantly better improvement in bone structural parameters, serum bone remodeling markers and bone mechanical strength compared to annatto T3 [29,41]. It should be noted that the dose of testosterone used was supraphysiological, thus partly explaining the more prominent skeletal effects [69].

Buserelin is a GnRH agonist commonly used in androgen ablation therapy [70]. A rat study showed that buserelin (subcutaneous, 75µg/kg bw/day, 3 months) induced testosterone deficiency and bone loss defined by deterioration in microstructure and strength comparable to, or even worse than, orchidectomy after treatment for three months [71]. The skeletal effects of annatto T3 (oral, 60 and 100 mg/kg bw/day, 3 months) and calcium supplement (1% in drinking water) on male rats treated with buserelin had been compared [30,31]. Annatto T3 at both doses increased proximal tibial bone volume and cortical thickness, as well as reduced trabecular separation. It also increased distal femoral bone volume and trabecular thickness [30,31]. They also increased osteoblast number without affecting osteoclast number [31]. Besides, both doses lowered serum C-terminal telopeptide of type 1 collagen (CTX-1) level but did not affect the osteocalcin level [31]. Considering the cellular results together, this may reflect that the bone resorption rate was reduced, but the bone formation rate was maintained. Annatto T3 also increased maximum load, stress, and elastic modulus of the bone, but only the dose of 60 mg/kg increased strain and calcein double-labelled surface [30,31]. The efficacy of annatto T3 was superior to calcium supplementation in many parameters tested. 

## 5. Effects of T3 on Bone Loss Due to Metabolic Syndrome

Metabolic syndrome is a collection of five medical conditions, including central obesity, hypertension, hyperglycemia, hypertriglyceridemia, and low high-density lipoprotein cholesterol, which present together, increase the risk of cardiovascular disease and diabetes mellitus exponentially [72]. The relationship between metabolic syndrome and osteoporosis is complicated because some components are harmful to bone health while others are protective [73]. This is reflected in the heterogeneity of the results obtained from previous epidemiological studies, whereby positive, negative and nil relationships between bone health and metabolic syndrome have been reported [74,75,76]. For instance, obesity is protective against osteoporosis because of mechanical loading and the expression of aromatase enzyme in adipose tissue, which produces estrogen peripherally. At the same time, chronic low-grade inflammation and bone marrow adiposity associated with obesity are detrimental to bone health [77]. Other components, such as increased calcium elimination due to hypertension [78], impaired osteoblast survival and function, as well as increased osteoclast formation due to oxidative stress associated with hyperglycemia [79,80,81] and oxidized lipoprotein [82,83,84], have a more straightforward, negative association with bone health. A previous study showed that metabolic syndrome induced by high-carbohydrate high-fat diet decreased bone volume, osteoblast number and osteoid surface but increased eroded surface and serum CTX-1 level in rats [85]. In another study, rats with metabolic syndrome induced by similar diet suffered from reduced tibial bone volume, trabecular number, connectivity density, cortical area, but increased trabecular separation [34]. However, the calcium content of tibial did not alter in these rats [34]. Biomechanically, the tibiae of these rats endured less load but higher strain and displacement [34]. Hence, this animal model is suitable as a bone loss model due to metabolic syndrome.

The rats fed with high-carbohydrate high-fat diet were supplemented with T3 from annatto and palm to assess changes that occurred to the skeletal system [32,33,39]. The supplementation was initiated 8 weeks after the diet was introduced to the rats. Researchers found that annatto T3 at 60 and 100 mg/kg bw/day for 16 weeks increased femoral bone volume, trabecular number, connectivity density and reduced trabecular separation and structural model index as assessed using micro-computed tomography [32]. Annatto T3 at both doses also increased osteoblast number and mineral apposition rate at the femur, implying that it could be anabolic to the bone [32]. Only annatto T3 at 100 mg/kg bw raised the load and Young’s modulus of the femur [32]. Another study used palm T3 at similar doses and showed that it prevented the decline in bone volume and trabecular number, as well as reducing structural model index and trabecular separation of the rats’ femurs [33]. Palm T3 at both doses also elevate load and Young’s modulus of the femur, but only the dose of 60 mg/kg bw improved femoral stiffness [33]. Similarly, palm T3 might be anabolic because it enhanced osteoblast surface and osteoid surface on the trabecular bone, but this was not reflected in the bone remodeling markers [39]. Both annatto T3 and palm T3 lowered the interleukin-1α and interleukin-6 [32,39]. 

## 6. Effects of T3 on Bone Loss Due to Cigarette-Smoking

Cigarette-smoking is recognized as an independent risk factor for bone loss and fracture [86,87]. In vitro studies showed that one of the major components of cigarette smoke, nicotine, decreased the formation of osteoblasts from human bone marrow mesenchymal stem cells [88]. Another study demonstrated that proliferation and formation of mineralized nodules were reduced in rat primary osteoblasts [89]. Gene expression analysis on these cells revealed two pathways related to bone metabolism affected by nicotine, i.e., Hedgehog and Notch pathways [89]. Other studies found that the effects of nicotine on osteogenesis might be bi-phasic in nature, whereby negative impacts were only observed with high doses [90]. Rats exposed to nicotine (intraperitoneal, 7 mg/kg/6 days a week, 4 months) showed decreased bone volume and trabecular number, bone mineralization and formation rate [91]. This might be contributed by increased bone resorption as shown through higher osteoclast number and eroded surface on the trabecular bone in the nicotine-treated rats compared to control [91]. The higher bone resorption was mediated by increased expression of proinflammatory cytokines, such as interleukin-1 and interleukin-6 [91]. The adverse effects of smoking on bone cannot be easily reversed even after cessation. This is illustrated by another animal, whereby the skeletal negative impacts of nicotine (intraperitoneal, 7 mg/kg/6 days a week, 2 months) could not be reversed after cessation for two months [92]. Changes in bone remodeling could be observed as early as 2 months, indicated by increased bone resorption marker (serum osteocalcin) and decreased bone resorption marker (serum pyridinoline), and they persisted after cessation [92]. 

Comparison of the effects of palm T3-enriched fraction, γ-T3 and α-TP in reversing bone damage due to exposure to nicotine has been attempted in male rats [42]. All treatment groups demonstrated improved bone volume, bone formation and reduction in osteoclast surface [42]. Groups treated with T3 experienced additional improvement in trabecular thickness, osteoblast number and eroded surface [42]. The mineralization rate and bone formation rate of rats receiving γ-T3 were higher compared to those receiving T3-enriched fraction and α-TP [42]. Further studies showed that T3-enriched fraction and γ-T3 prevented nicotine-induced interleukin-1 and interleukin-6, as well as halting the increase in serum pyridinoline [43]. Serum osteocalcin was also increased with both treatments [43]. These changes were not seen with α-TP treatment [43]. A gene expression study showed that palm vitamin E (oral, 60 mg/kg/day, 2 months) rescued suppression of RUNX2, OSX and BMP-2 mRNA expression in the femur of rats post-nicotine administration (intraperitoneal, 7 mg/kg/6 days a week, 2 months) in male rats [93]. 

## 7. Effects of T3 on Bone Loss Due to Alcohol

Epidemiological studies have shown that light alcohol intake may be beneficial to the bone but heavy alcohol use is associated with bone loss in men [94]. The mechanism of alcohol-induced bone loss is complex [95]. Male alcohol abusers were reported to have a lower testosterone level and higher oxidative stress compared to healthy control [96]. Chronic alcohol use also increases the production of proinflammatory cytokines, such as tumor necrosis factor-α and interleukin-1β, which will impair bone formation [97]. Besides, high-dose alcohol was found to regulate mammalian target of rapamycin (mTOR) pathway and reduce osteoblast and bone formation (indicated by reduced runt-related factor-2 (RUNX2) and ALP expression) and increase adipocyte formation (indicated by increased peroxisome proliferator-activated receptor-γ expression) in the bone marrow [98]. Serum sclerostin level was reported to increase in alcoholic patients, and it correlated negatively with serum osteocalcin (a bone formation marker) and positively with telopeptide (a bone resorption marker) [99]. Other bone related pathways that may be impacted by chronic alcohol use include vitamin D-parathyroid (PTH) axis and insulin-like growth factor-1 (IGF1) - growth hormone (GH) signaling [95]. 

A previous study compared the bone-sparing effects of palm vitamin E and α-TP (oral, 60 mg/kg bw/day, 2 months after the last alcohol ingestion) in a binge-drinking male rat model (oral, 3 g/kg bw 20% ethanol in saline, 3 days a week for 4 weeks). Both palm vitamin E and α-TP increased tibial calcium and magnesium compared to the vehicle-treated group. Palm vitamin E but not α-TP was shown to improve the biomechanical strength of the tibiae as evidenced by increased maximum force, ultimate stress and Young’s modulus values. However, the authors did not investigate the mechanism of action of palm vitamin E in reducing the skeletal damage of alcohol but suggested the antioxidative and anti-inflammatory properties of T3 might play a role [38]. 

## 8. Effects of T3 on Bone Loss Due to Glucocorticoid

Long-term glucocorticoid use is a major risk factor of osteoporosis. Glucocorticoid mainly affects bone formation, whereby it impairs the differentiation and function of osteoblasts [100]. High-dose glucocorticoid reduces the expression of Wnt and increases the expression of Wnt inhibitors secreted frizzled-related protein and dickkopf-related protein 1 (DKK1) in mature osteoblasts [101]. It also increases serum sclerostin level that correlates negatively with bone formation markers [102]. Glucocorticoid activates glycogen synthase kinase-3β in the Wnt-signaling pathway, thereby increases the phosphorylation of β-catenin by GSK-3 and inhibits nuclear translocation of β-catenin [103]. It also antagonizes Runx2 signaling and reduces the expression of several genes related to osteoblastogenesis [104]. The effects of glucocorticoid on osteoclastogenesis are mainly a derivative of decreased osteoprotegerin (OPG) production by osteocytes, which promotes osteoclastogenesis [105]. Indirectly, glucocorticoid reduces blood testosterone level and calcium absorption, as well as altering IGF-1-GH axis, which altogether cause bone loss [106,107,108]. 

The effects of T3 in preventing bone loss due to glucocorticoid have been explored in a few studies. Male rats were adrenalectomized and supplemented with dexamethasone to mimic glucocorticoid replacement therapy in human post-adrenalectomy. γ-T3 (oral, 60 mg/kg bw/day for 8 weeks) was shown to preserve bone lumbar calcium content in rats given low (intramuscular, 120 mg/kg/bw) and high-dose dexamethasone (intramuscular, 240 mg/kg/bw), although it did not affect the BMD of the rats [37]. It also prevented the dexamethasone-induced increase in fat mass assessed by DXA. α-TP at the same dose did not demonstrate similar effects in the same study [37]. In another study, dexamethasone (intramuscular, 120 mg/kg/bw) reduced BMD gained in 8 weeks, femoral length and calcium content [36]. Palm vitamin E (oral, 60 mg/kg bw/day for 8 weeks) prevented all these changes [36]. 

A summary of the bone-sparing properties of T3 is presented in Table 2. 

## 9. Mechanism of Action of T3 in Protecting Bone Health

### 9.1. Oxidative Stress

The role of oxidative stress in bone remodeling has been established in preclinical studies. Free radical species, especially hydrogen peroxide, facilitate intracellular signaling in pre-osteoclasts and promote their differentiation into mature osteoclasts and bone resorption activity through nuclear factor-κB (NF-κB) and mitogen-activated protein kinase (MAPK) pathways [51]. Oxidative stress also decreases the survival of osteoblasts and osteocytes, encourages their apoptosis and diminishes bone formation [109]. However, the relationship between oxidative stress and bone is complicated in vivo. Smoking [110], alcohol drinking [111] and metabolic syndrome [112] are contributors to oxidative stress in vivo regardless of sex. Estrogen is regarded as an antioxidant and postmenopausal osteoporosis is suggested to be partly attributable to increased oxidative stress [113]. On the other hand, the role of androgen on oxidative stress is controversial, whereby some studies reported a protective role of androgen against oxidative stress [114,115,116] while others demonstrated that androgen further enhanced oxidative stress in various systems [117,118,119]. Currently, the contribution of oxidative stress in bone loss due to androgen deficiency is uncertain. 

In vitro studies showed that oxidative stress induced by hydrogen peroxide on primary osteoblasts overwhelmed their antioxidative enzyme defense. This subsequently promotes the apoptosis of these osteoblasts. γ-T3 prevented the decrease in the activity of antioxidative enzymes and apoptosis of osteoblasts [120]. However, high-dose γ-T3 was demonstrated to be cytotoxic to osteoblasts [50,120]. In contrast, α-TP did not prevent the apoptotic effects of hydrogen peroxide in osteoblasts [50]. It is not known whether T3 achieved its protective activities by acting as a free radical scavenging agent or by regulating the NF-E2-related factor 2 (NRF2)-associated antioxidant responsive element in the cells. A previous study demonstrated that γ-T3 was able to stabilize the expression of NRF2 in keratinocytes and promoted the mRNA expression of heme oxygenase-1 and NAD(P)H:quinone oxidoreductase-1 [121]. Hence, the mechanism of antioxidant action on T3 in protecting bone cells against oxidative damage should be examined further.

### 9.2. Inflammation

The NF-κB pathway plays a very significant role in bone remodeling. It is a family of transcription factors involved in inflammatory and immune response [122]. The pathway can be divided into canonical and non-canonical. In the canonical pathway, IKB kinase (IKK), upon activation by stimuli, will phosphorylate IKBα and subject it to degradation, subsequently enable nuclear translocation of NF-κB [122]. Activation of NF-κB pathway leads to transcription of genes related to inflammation, including cytokines, cell adhesion molecules and chemokines [123]. IKKβ (a component of IKK) is critical in the survival and differentiation of osteoclast progenitors [124]. It prevented apoptosis of osteoclast progenitors induced by Jun N-terminal kinase (JNK) activation [124]. Deletion of IKKβ caused osteopetrosis in mice due to defective osteoclast formation [125]. IKKβ also rescued the osteoclast progenitors from tumor necrosis factor (TNF) α-induced apoptosis [125]. Deletion of IKKβ also prevented endotoxin-induced inflammatory bone loss in mice [125]. Activation of the non-canonical pathway promotes TNF-induced osteoclast formation through increased RelB. Inhibition of TNF receptor-associated receptor (TRAF) 3 and p100 also reduced osteoclastogenesis [126,127]. With regards to bone formation, inhibition of NF-κB through IKK mutation stimulates osteoblast function but not its differentiation [128]. Ovariectomized mice with mutated IKK were protected against bone loss. These effects were mediated through up-regulation of JNK/Fra-1 pathway [128]. The regulation of NF-κB by various stimuli would explain the pathogenesis of bone loss secondary to inflammatory diseases. 

T3 could suppress the activation of NFκB pathway, as evidenced from previous studies using cancer cells or macrophages [129,130,131]. Particularly, γ-T3 and δ-T3 suppressed NF-κB activity better than other T3 isomers in a study using pancreatic cells [132]. T3 inhibited IKK activation, phosphorylation and degradation of IKBα and prevented nuclear translocation of NF-κB [130,131,133]. γ-T3 was shown to block NF-κB reporter gene transcription induced by TNF, TRAF2 and NF-κB-inducing kinase. This demonstrated that T3 can block the non-canonical pathway [133]. T3 was shown to lower inflammatory cytokines levels in an animal model of osteoporosis induced by metabolic syndrome [32,33], nicotine [43] and ferric nitrilotriacetate [134]. However, there is no direct evidence demonstrating the suppression of NF-κB pathway by T3 in bone cells. The closest evidence is the suppression of LPS-induced IKBα in the bone marrow-derived macrophages, which could serve as precursors to osteoclasts, by γ-T3 [130]. This could explain the inhibitory effects of γ-T3 on osteoclast formation from bone marrow macrophages in a separate study [135]. 

### 9.3. Mevalonate Pathway

The mevalonate pathway, known to be responsible for cholesterol synthesis, also produces isoprenoids involved in prenylation of signaling proteins, known as GTPases [136]. These proteins are known to regulate bone remodeling [137]. For instance, PTH-induced osteoblastogenesis was shown to be mediated by suppression of the mevalonate pathway and Rho-associated protein kinase inhibition [138]. Osteoclast functions, including the formation of ruffled border and sealing zone, depend on RhoA, Rac, Cdc42, RhoU, and Arf6. Ras regulates the survival of osteoclasts [139]. Nitrogen-containing bisphosphonates, a class of antiosteoporosis agents, exert their functions by inhibiting farnesyl-diphosphate synthase [140]. Statins, a class of cholesterol-lowering agents, inhibits 3-hydroxy-3-methyl-glutaryl-CoA reductase (HMGR), the rate-determining enzyme of the mevalonate pathway to achieve its therapeutic effects [141]. Although statins are not used in the treatment of osteoporosis, its prolonged consumption has been associated with increased hip and lumbar BMD and reduced hip fracture risk in humans [142]. Several preclinical studies have established that T3 is a potent suppressor of the mevalonate pathway. T3 was shown to decrease the protein expression of HMGR post-transcriptionally by increasing its degradation rate in hepatocytes [143]. This was echoed by studies using cancer cells, demonstrating down-regulation of HMGR expression by T3, which partially explains its anticancer activities [144,145,146]. As a result, of HMGR suppression, metabolites along the mevalonate pathway, such as farnesyl-diphosphate, squalene and cholesterol were reduced [147]. 

The involvement of mevalonate pathway in the bone-sparing effects of T3 has been implied in several studies. Studies in ovariectomized rats showed that supplementation of annatto T3 in combination with lovastatin prevented bone loss better than individual treatments, as evaluated through bone histomorphometry, bone calcium content and bone strength [26,27,28]. The combination therapy also enhanced the mRNA expression of BMP-2 in the femur of the rats better than annatto T3 or lovastatin alone [28]. In an ovariectomized mice model, daily supplementation of mevalonate (oral, 25 mg/kg bw, daily for three months) diminished the bone-protective effects of γ-T3 (intraperitoneal, 100 mg/kg bw, once monthly for three months) as evaluated by bone structural and histomorphometric indices [148]. Mevalonate also suppressed γ-T3 induced elevation of transcription factors mRNA expression for osteoblastogenesis, such as osterix and RUNX2. Up-regulation of OPG mRNA and down-regulation of RANKL mRNA by γ-T3 in mice was abolished by mevalonate co-treatment [148]. The observation was validated in an in vitro study using bone marrow cells and UAMS-32P cells. Treatment with γ-T3 significantly reduced the intracellular farnesyl pyrophosphate and geranylgeranyl pyrophosphate level, both of which are intermediates in the mevalonate pathway. γ-T3 mediated suppression of osteoclast-like cell formation stimulated by parathyroid was also abrogated by mevalonate [148]. 

A summary of the suggested mechanism of actions of T3 in preventing bone loss is presented in Figure 2. 

## 10. Perspectives on the Use of T3

Three major concerns on the application of T3 as a bone-sparing agent are its safety, bioavailability, and marketability. In terms of safety, one study in postmenopausal women with osteopenia revealed that annatto T3 at 600 mg for 12 weeks did not affect their liver and kidney functions [149]. It also suppressed the high turnover markers among the subjects, indicated by reduced serum bone ALP and urinary N-terminal telopeptide. This was accompanied by a reduced level of 8-hydroxy-2’-deoxyguanosine, an oxidative stress marker [150]. However, similar studies on the use of T3 for bone protection in men is absent. Toxicity studies in female mice revealed increased bleeding risk with high-dose palm vitamin E (oral, 500 and 1000 mg/kg for 14 and 42 days) [151], but this dose was much higher than the desired dose for bone strengthening effects. Similar toxicity studies in male animals or men were limited. 

In terms of bioavailability, T3 generally showed much lower bioavailability compared to α-TP due to the competitive binding at α-TP transport protein, which regulates the circulating vitamin E level [21]. However, a previous study established that a single-dose of emulsified γ-T3 was deposited in the rat femur and spine 14 days after supplementation [152]. The treatment also exerts biological effects on the bone, by up-regulating OPG mRNA and down-regulating RANKL mRNA [152]. Various means to increase the bioavailability of T3 are being studied, including the use of the self-emulsifying system [153] and nanoparticles [154]. 

There are some commercial challenges in developing T3 as a pharmaceutical product to prevent osteoporosis in men. A patent search revealed two relevant patents pertaining to the use of T3 as an antiosteoporotic agent [155,156]. However, the T3 supplements are prevalent in the market, and natural composition is usually used. It might hamper the interest of pharmaceutical companies to invest and develop it as pharmaceutical products. However, it still has great potential to be developed into functional foods and nutraceuticals for men to halt bone loss. In addition, T3 also showed beneficial effects against other age-related diseases, such as metabolic syndrome [157,158], neurogenerative disease [159], arthritis [160,161], sarcopenia [162,163], which may be attractive to elderly men suffering from multiple conditions concurrently.

## 11. Limitations

This review is not without its limitations. The ultimate consequence of male osteoporosis is fragility fracture, which carries significant morbidity and mortality to the patients. Fragility fracture is not only predicted by reduced bone mass, but also muscle and cognitive functions of an individual, which are related to the gait and tendency to fall [164]. This review has only addressed the effects of T3 on bone mass, structure, and strength, but not on evidence for muscle and cognitive functions. However, osteoporosis remains the most treatable predictors of fracture, and it can be prevented by T3 supplementation. 

## 12. Conclusions

As a conclusion, evidence accumulated thus far has demonstrated promising effects of T3 as a preventive agent against male osteoporosis in models of bone loss induced by testosterone deficiency (surgical or chemical ablation), metabolic syndrome, cigarette-smoking, glucocorticoids and alcoholism. Men with these risk factors or osteopenia could benefit from T3 supplementation to stop the progression of bone loss. The human equivalent dose of T3 proven to prevent osteoporosis is approximately 600 mg/day, to be taken after meals to enhance absorption. However, there is no clinical trial to study the skeletal effects of T3 in men so far. Thus, the efficacy of T3 in preventing the progression of male osteoporosis still awaits new clinical evidence in humans. The bone-sparing effects of T3 could be mediated by its antioxidant, anti-inflammatory, and mevalonate-suppressive effects. However, more studies are needed to illustrate its mechanism of actions, especially the cell signaling pathways involved. 

## Figures and Tables

**Figure 1 ijms-20-01355-f001:**
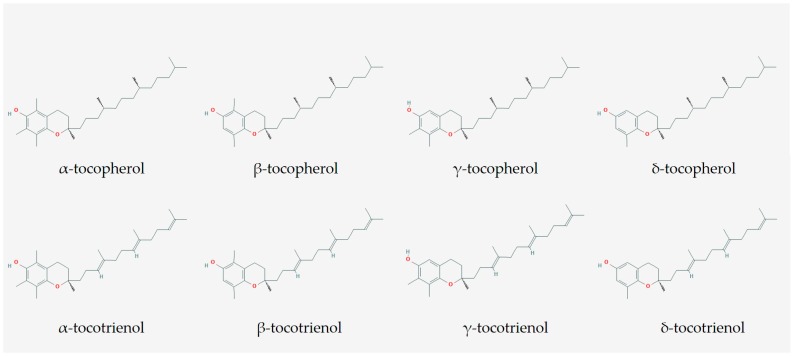
The molecular structure of tocopherol (TF) and tocotrienol (T3). The images are obtained from https://pubchem.ncbi.nlm.nih.gov/.

**Figure 2 ijms-20-01355-f002:**
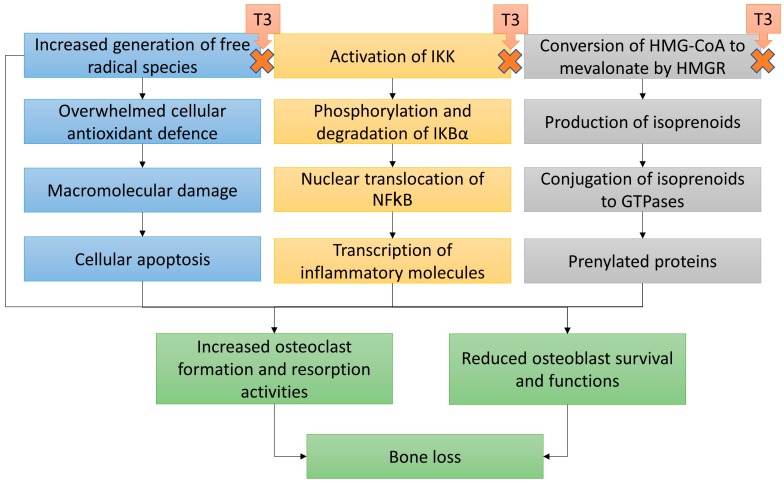
The proposed mechanism of action of tocotrienol (T3) in protecting bone health. Abbreviation: HMG-CoA, 3-hydroxy-3-methylglutaryl-CoA; HMGR, 3-hydroxy-3-methylglutaryl-CoA reductase; IKB, nuclear factor of kappa light polypeptide gene enhancer in B-cells inhibitor, alpha; IKK, IKB kinase; NF-κB, nuclear factor kappa-light-chain-enhancer of activated B-cells.

**Table 1 ijms-20-01355-t001:** The composition of tocotrienol (T3) mixture used in experiments

Reference	Vitamin E Used	Composition of Vitamin E (%)			
		αTP	αT3	γT3	δT3
[35]	Palm vitamin E	24.4	21.6	27.7	11
[36,37]	Palm vitamin E	24.83	20.73	26.68	13.32
[38]	Palm vitamin E	22.48	23.16	36.89	12.57
[33,39]	Palm vitamin E	21.9	24.7	36.9	12
[46]	Palm T3	18.43	14.62	32.45	23.93
[29,30,31,32,40,41]	Annatto T3			10	90
[42,43]	Palm T3-enriched fraction		43	31	14

**Table 2 ijms-20-01355-t002:** Skeletal properties affected by tocotrienol (T3).

Ref	Induction of Bone Loss	Treatment	Period	Skeletal Properties Affected by T3																			
				BV/TV	Tb.N	Tb.Th	Tb.Sp	SMI	Connectivity Density	Cortical Indices	Ob.N or Ob.S	Oc.N or Oc.S	ES/BS	OS/BS	OV/BV	sLS/BS	dLS/BS	MS	MAR	BFS/MS	BMD	Bone Calcium Content	Biomechanical Strength
[29]	Orchidectomy	AnT3 60 mg/kg	2 months	↑	↑	↔	↓	↔	↔							↓	↑	↔	↔	↔			
[40]	Orchidectomy	AnT3 60 mg/kg	2 months								↑	↓	↓	↑	↑								
[41]	Orchidectomy	AnT3 60 mg/kg	2 months																			↑ (tibia)	↔
[35]	Orchidectomy	PVE 30 mg/kg	8 months																		↑	↑ (lumbar)	
[30]	Chemical castration by buserelin	AnT3 60 or 100 mg/kg	3 months	↑	↑	↔	↓	↔	↑	↑ thickness												↑ (femur)	↑
[31]	Chemical castration by buserelin	AnT3 60 or 100 mg/kg	3 months	↑	↔	↑	↔					↓	↔	↔	↔	↓	↑ (60 mg/kg only)	↔	↔	↔			
[32]	Metabolic syndrome	AnT3 60 or 100 mg/kg	4 months	↑	↑	↔	↓	↓	↑	↔	↑	↔	↔	↔	↔	↑ (60 mg/kg)	↔	↔	↑ (60 mg/kg only)	↔		↔ (femoral)	↑
[33]	Metabolic syndrome	Palm T3 60 or 100 mg/kg	4 months	↑	↑	↔	↓	↓	↔	↔												↔ (femoral)	↑
[39]	Metabolic syndrome	Palm T3 60 or 100 mg/kg	4 months	↑	↔	↔	↔				↑	↔	↓ (100 mg/kg)	↑	↔	↓ (60 mg/kg)	↔	↔	↔	↔	↔		
[37]	Glucocorticoid	γ-T3 60 mg/kg	2 months																		↔	↑ (lumbar)	
[36]	Glucocorticoid	PVE 60 mg/kg	2 months																		↑	↑ (femoral)	
[42]	Nicotine	Palm T3 enriched fraction, γ-T3 60 mg/kg	2 months	↑	↔	↑					↑	↓	↓			↓			↑	↑			
[38]	Alcohol	PVE 60 mg/kg	2 months																			↑ (tibial)	↑

Abbreviation: BV/TV, bone volume; Tb.N, trabecular number; Tb.Th, trabecular thickness; Tb.Sp, trabecular separation; SMI, structural model index; Ob.N. or Ob.S, osteoblast number or surface; Oc.N or Oc.S, osteoclast number or surface; ES/BS, eroded surface; OS/BS, osteoid surface; OV/BV, osteoid volume; sLS/BS, single-labelled surface; dLS/BS, double-labelled surface; MS, mineralizing surface; MAR, mineral apposition rate; BFS/MS, bone formation rate; BMD, bone mineral density; AnT3, annatto tocotrienol; PVE, palm vitamin E; ↑, increase/improve; ↓ reduce; ↔ no change.

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
