# Peer review of "The Role of Tocotrienol in Preventing Male Osteoporosis—A Review of Current Evidence"

_ijms, 2019, doi:10.3390/ijms20061355_

Reviewer 1 Report

The authors have done a nice job assessing the effect of tocotrienol in male osteoporosis; however, there are some issues that need to be addressed before the article could be published.

- First, the article needs to be edited for English as there are several grammatical mistakes, mainly about the used propositions and in some phases the correct tense of the verbs are not used.

- For me the title of the article is kind of misleading. The article contains mixed information about animal and human studies, so I dont feel using "evidence from preclinical model" is correct. Moreover, the title shows the focus of the article is on male osteoporosis, which is kind of correct but based on the title the reader expect the conclusion to be on male osteoporosis and contain recommendations in this regard. This is while the conclusion in quite broad with no mention of male osteoporosis. 

- The section, "the significance of bone parameters" does not really fit in this article. The results of the reported studies do not focus on BMD or bone histomorphometry results so I dont feel the need for this section.

_ In my opinion it would also make more sense to first discuss the mechanisms and then the effect on bone loss due to various risk factors. Moreover, the fact that for some risk factors the results are from animal studies and for the others human studies is also confusing. The authors should also mention why they have selected these risk factors from among the long list of risk factors linked with osteoporosis

- As mentioned earlier, the conclusion needs to be rewritten. Based on the topic, the conclusion should at least contain some kind of recommendation for T3 usage and also should focus on male osteoporosis

Author Response

Manuscript Title: The Role of Tocotrienol in Preventing Male Osteoporosis – A Review of Current Evidence

Manuscript ID: ijms-453207

Thank you for reviewing our manuscript and your constructive comments.

Comment

Reply

First,   the article needs to be edited for English as there are several grammatical   mistakes, mainly about the used propositions and in some phases the correct   tense of the verbs are not used.

Thank   you for the suggestion. We have invited a native speaker to proofread the   manuscript and put it through a language editing software.

For me the title of the article is kind of   misleading. The article contains mixed information about animal and human studies, so I don’t feel   using "evidence from preclinical model" is correct. Moreover, the   title shows the focus of the article is on male osteoporosis, which is kind   of correct but based on the title the reader expect the conclusion to be on   male osteoporosis and contain recommendations in this regard. This is while   the conclusion in quite broad with no   mention of male osteoporosis.

Thank   you for the suggestion. We have deleted “evidence from preclinical model”   from the title.

We have   also amended the conclusion so that it is more relevant to male osteoporosis.  

The   section, "the significance of bone parameters" does not really fit   in this article. The results of the reported studies do not focus on BMD or   bone histomorphometry results so I dont   feel the need for this section.

Thank   you for the suggestion. We have removed the section as per your suggestion.

In my opinion it would also make more sense to   first discuss the mechanisms and then the effect on bone loss due to various risk   factors. Moreover, the fact that for some risk factors the results are from   animal studies and for the others human   studies is also confusing. The authors   should also mention why they have selected these risk factors from among the   long list of risk factors linked with osteoporosis

Thank   you for the suggestion. We have added the mechanisms and risk factors of bone   loss in the introduction (2nd paragraph).

The   risk factors elaborated in each section are based on the models of bone loss   which tocotrienols have been tested on (as stated in the last sentence of   introduction).

In   each section, the relationship between risk factors and osteoporosis is first   explained using animal and human studies. Then, the effects of tocotrienol as   illustrated using relevant models of bone loss are highlighted.

As   mentioned earlier, the conclusion needs to be rewritten. Based on the topic,   the conclusion should at least contain some kind of recommendation for T3   usage and also should focus on male osteoporosis

Thank   you for the suggestion. We have rectified the conclusion so that it is more   relevant to male osteoporosis.

As a conclusion, evidence accumulated thus   far has demonstrated promising effects of T3 as a preventive agent against   male osteoporosis in models of bone loss induced by testosterone deficiency   (surgical or chemical ablation), metabolic syndrome, cigarette-smoking and   alcoholism. Men with these risk factors or osteopenia could benefit from T3 supplementation to stop the   progression of bone loss. The human equivalent dose of T3 proven to prevent   osteoporosis is approximately 600 mg/day, to be taken after meals to enhance   absorption. The bone-sparing effects of T3 could be mediated by its   antioxidant, anti-inflammatory and mevalonate-suppressive effects. However,   more studies are needed to illustrate its mechanism of actions and its   effects on men suffering from   osteopenia and osteoporosis.”

Reviewer 2 Report

This review by Kok-Yong Chin and Loelaiman Ima Nirwana intends in describing bone protective effects of tocotrienol in male osteoporosis by summarizing the data obtained rodent osteoporosis models. It is suggested that the administration of tocotrienol improves bone property in the osteoporosis models. Also, they described the mechanisms; antioxidant, anti-inflammatory, and mevaloneate-suppressive pathways.

Overall, the manuscript is well written by referring their works and other reports. It could still be refined for better impact.

1)The author should refer to LOH (late-onset hypogonadism) syndrome in section 5. In this setting of androgen deficiency, male osteoporosis is one of the main features of the LOH syndrome. The concept of LOH syndrome is gathering attention. By referring the concept, the clinical impact of this section will be strengthened.  

2)The author should include tables to summarize main findings of each models.

  Though the authors described main findings of each model at each section, it’s hard to follow, because the measurement parameters which the authors introduced are inconsistence among sentences. Tabular forms should be prepared by referring main measurement parameters, e.g. BV/TV, TbN, TbTh, ObN, OcN, ES/BS, as the author published previously in table 2 of Drug Des Devel Ther. 2015 Apr 8;9:2049-61.

3)Structures of tocotrienol and tropherol could be provided.

Minor points

1)Abbreviation should be used in the following parts once the abbreviation is defined, e.g. tocotrienol at Page 2, lines 59, 62, and 95, and in table 1  

Author Response

Manuscript Title: The Role of Tocotrienol in Preventing Male Osteoporosis – A Review of Current Evidence

Manuscript ID: ijms-453207

Thank you for reviewing our manuscript and your constructive comments.

Comment

Reply

The   author should refer to LOH (late-onset hypogonadism) syndrome in section 5.   In this setting of androgen deficiency, male osteoporosis is one of the main   features of the LOH syndrome. The concept of LOH syndrome is gathering   attention. By referring the concept, the clinical impact of this section will   be strengthened. 

Thank   you for the suggestion. We have added some descriptions on late-onset   hypogonadism in men in Section 4 (renumbered).

The   author should include tables to summarize main findings of each models.

Thank   you for the suggestion. The main findings for each model have been summarized   in Table 2.

Though   the authors described main findings of each model at each section, it’s hard   to follow, because the measurement parameters which the authors introduced   are inconsistence among sentences. Tabular forms should be prepared by   referring main measurement parameters, e.g. BV/TV, TbN, TbTh, ObN, OcN,   ES/BS, as the author published previously in table 2 of Drug Des Devel Ther.   2015 Apr 8;9:2049-61.

Thank   you for your suggestion. We have added a similar table as Table 2.

Structures   of tocotrienol and tocopherol could be provided.

Thank   you for the suggestion. We have added the molecular structure of tocopherol   and tocotrienol as Figure 1.

Abbreviation   should be used in the following parts once the abbreviation is defined, e.g.   tocotrienol at Page 2, lines 59, 62, and 95, and in table 1 

Thank   you for the reminder. We have changed “tocotrienol” to T3 in the manuscript.

Round  2

Reviewer 1 Report

The authors have done a great job revising the article but there are still some parts that need to be revised. First of all the classification for the male osteoporosis is not completely correct. This should be divided into primary and secondary osteoporosis and then metabolic syndrome and .... are causes of the secondary osteoporosis and it should be mentioned that there are also other conditions leading to osteoporosis in this regard which are not mentioned here. Another question is that while reading the text I had the impression that authors have suggested that smoking cessation would lead to osteoporosis which is not true. Smoking is the risk factor.

The conclusion needs to be rewritten. For instance the first line the concerns for using T3 is an important concern and so could have a dedicated section to be discussed. So most of the conclusion would be directed to that section and then the authors need to provide a more descriptive conclusion for their article. Adding a limitation section mentioning lack of information regarding T3 in different aspects would also be helpful

Author Response

Manuscript title: The Role of Tocotrienol in Preventing Male Osteoporosis – A Review of Current Evidence

Manuscript ID: ijms-453207

Comment

Reply

The   authors have done a great job revising the article but there are still some   parts that need to be revised. First of all the classification for the male   osteoporosis is not completely correct. This should be divided into primary   and secondary osteoporosis and then metabolic syndrome and .... are causes of   the secondary osteoporosis and it should be mentioned that there are also   other conditions leading to osteoporosis in this regard which are not mentioned here.

Thank   you for the suggestion. We have modified the 2nd paragraph of Introduction to   describe the types of male osteoporosis and its risk factors.

Male osteoporosis can be classified into   primary and secondary osteoporosis. The cause of primary male osteoporosis is   age-related bone loss (senile osteoporosis)   or unknown (idiopathic osteoporosis). Secondary male osteoporosis is caused   by lifestyles, medical conditions or medications harmful to the bone. Some of   the lifestyle behaviours contributing to male osteoporosis are excessive   consumption of alcohol and caffeinated beverages, cigarette-smoking and   physical inactivity. Male osteoporosis can occur secondary to other diseases,   such as hypogonadism, gastrointestinal disease, hyperparathyroidism and   thyrotoxicosis. Prolonged use for medications, such as glucocorticoids,   antineoplastic agents and anticonvulsants, will also cause osteoporosis

Another   question is that while reading the text I had the impression that authors   have suggested that smoking cessation would lead to osteoporosis which is not   true. Smoking is the risk factor.

Thank   you for the comment. We acknowledge that smoking the risk factor of male   osteoporosis. This has been clearly established at the start of section 6: “Cigarette   smoking is recognized as an independent risk factor for bone loss and   fracture

We   also want to alert the readers that the   negative impact of smoking on bone is   not easily reversible. Thus, towards the end of the 1st paragraph   in section 6, we mentioned that

The adverse effects of smoking on bone cannot be easily reversed even after   cessation. This is illustrated by another animal, whereby the skeletal   negative impacts of nicotine (intraperitoneal, 7 mg/kg/6 days a week, 2   months) could not be reversed after cessation for two months…

The   conclusion needs to be rewritten. For instance   the first line the concerns for using T3 is an important concern and so could   have a dedicated section to be discussed. So most of the conclusion would be   directed to that section and then the authors need to provide a more   descriptive conclusion for their article. Adding a limitation section   mentioning lack of information   regarding T3 in different aspects would also be helpful

Thank   you for the comments.

We   have rearranged the “Conclusion and Perspectives” into three new sections.

‘Section   10 - Perspectives on the use of T3’ discussed practical issues on the   development and use of T3 as bone health supplements.

‘Section   11 - Limitations’ discussed some important aspects of fragility fracture not   reviewed in this paper.

‘Section   12 – Conclusion’ is the descriptive conclusion for this review.

Round  3

Reviewer 1 Report

The authors have perfectly addressed the comments. Though an additional phrase mentioning that in this article some of the causes of secondary osteoporosis including .... would be discussed and the reason only these causes have been selected is ....